# Effect of Melatonin on Herpesvirus Type 1 Replication

**DOI:** 10.3390/ijms25074037

**Published:** 2024-04-04

**Authors:** Zulema Pérez-Martínez, Jose Antonio Boga, Yaiza Potes, Santiago Melón, Ana Coto-Montes

**Affiliations:** 1Servicio de Microbiología, Hospital Universitario Central de Asturias, 33011 Oviedo, Spain; zperezmrtnz@gmail.com (Z.P.-M.); joseantonio.boga@sespa.es (J.A.B.); santiago.melon@sespa.es (S.M.); 2Instituto de Investigación Sanitaria del Principado de Asturias (ISPA), 33011 Oviedo, Spain; potesyaiza@uniovi.es; 3Instituto de Neurociencias del Principado de Asturias (INEUROPA), 33006 Oviedo, Spain; 4Departamento de Morfología y Biología Celular, Universidad de Oviedo, 33006 Oviedo, Spain

**Keywords:** melatonin, herpesvirus, oxidative stress, ER stress, autophagy, apoptosis

## Abstract

Acute HSV-1 infection is associated with mild symptoms, such as fever and lesions of the mouth, face and skin. This phase is followed by a latency period before reactivation, which is associated with symptoms ranging from ulcers to encephalitis. Despite available anti-HSV-1 drugs, the development of new antiviral agents is sought due to the presence of resistant viruses. Melatonin, a molecule secreted by the pineal gland, has been shown to be an antioxidant, inducer of antioxidant enzymes, and regulator of various biological processes. Clinical trials have explored its therapeutic utility in conditions including infections. This study focuses on melatonin’s role in HSV-1 replication and the underlying mechanisms. Melatonin was found to decrease the synthesis of HSV-1 proteins in infected Vero cells measured by immunofluorescence, indicating an inhibition of HSV-1 replication. Additionally, it regulates the activities of antioxidant enzymes and affects proteasome activity. Melatonin activates the unfolded protein response (UPR) and autophagy and suppresses apoptosis in HSV-1-infected cells. In summary, melatonin demonstrates an inhibitory role in HSV-1 replication by modulating various cellular responses, suggesting its potential utility in the treatment of viral infections.

## 1. Introduction

Infection with herpes simplex virus type 1 (HSV-1) is typically characterized by three distinct phases that correspond to the clinical course experienced by most patients: acute infection, latency, and reactivation. In most cases, acute infection occurs during early childhood and is often asymptomatic or may present with only mild symptoms such as fever and rash. During primary infection, viruses are transported via sensory ganglia to establish a chronic latent infection, most commonly in the trigeminal, cervical, or lumbosacral ganglia. Clinically evident reactivation, often in the setting of immune compromise, can be associated with different pathologies such as cold sores, herpes stromal keratitis, eczema herpeticum, meningitis and herpes simplex encephalitis [1]. The majority of anti-HSV drugs are nucleoside analogues such as acyclovir and penciclovir that directly target the viral DNA polymerase when in their phosphorylated form. Since the first phosphorylation of antiviral nucleoside analogues typically occurs via the viral enzyme thymidine kinase (TK), the main basis for resistance resides in mutations in this viral enzyme. One way to overcome this requirement for TK is to use a nucleotide analogy that already has a monophosphate attached, like cidofovirand adefovir or a non-nucleotide analogue such as foscarnet [2]. In any case, acyclovir is the most used antiherpetic and the prevalence of acyclovir-resistant isolates among immunocompetent and immunocompromised hosts has been reported at approximately 0.3–5% and 4–7%, respectively [3]. Gaining knowledge of the cellular mechanisms that the virus alters could provide interesting avenues to detect targets and design new antiviral strategies. The absence of viral machinery for protein synthesis, which makes the virus completely dependent on cellular machinery for the production of its own proteins, makes the endoplasmic reticulum, as the cell’s main protein producer, a prime candidate for study. Thus, the new viral demand for proteins will induce an excess of endoplasmic request that will result in an overproduction of misfolded proteins known as endoplasmic reticulum stress [4], which usually exceeds the capacity of the proteasome. To try to counteract this stress, the cell develops its multifactorial misfolded protein response system (known as unfolded protein response, UPR), which triggers consecutive response cascades depending on the severity of the induced stress [5] and may ultimately induce the activation of survival mechanisms, such as autophagy or programmed cell death or apoptosis. Melatonin (N-acetyl-5-methoxytryptamine) is a highly conserved multifunctional signalling molecule. On the one hand, melatonin is a hormone secreted in circadian rhythm by the pineal gland, with a systemic light/dark information function (low daytime levels and nocturnal peak) [6]. However, this indolamine is also produced at constant levels in most cells of the organism, acting as a potent antioxidant and inducer of antioxidant enzymes, and modulator of endoplasmic reticulum stress, as well as regulator of apoptosis and autophagy. These functions may explain its role in various biological processes [7,8]. Indeed, numerous clinical trials have examined the therapeutic usefulness of melatonin in preventing cell damage in both acute (sepsis, asphyxia in newborns) [9] and chronic (metabolic and neurodegenerative diseases, cancer, inflammation, aging) states [10]. Their role has been also studied in infections, especially in those caused by viruses, which often result in increased oxidative stress and inflammatory injury [11]. In fact, some evidence of the ability of melatonin to protect against viral infections has been reported. Encephalomyocarditis virus (EMCV) is a highly virulent and aggressive pathogen that triggers inflammation in the brain and heart of rodents. When mice were infected with non-lethal doses of EMCV, the administration of melatonin prevented paralysis and fatalities [12]. Melatonin also demonstrates protective properties in mice infected with Semliki Forest virus (SFV), an arbovirus known for causing encephalitis by infiltrating the central nervous system (CNS) and eventually leading to death. Injected melatonin not only decreased mortality rates but also significantly delayed the onset of symptoms and eventual death [13]. Despite the fact that attenuated West Nile virus (WN-25) is not neuroinvasive and does not typically induce encephalitis, exposure to stressors can trigger WN-25 encephalitis in mice. Melatonin can counteract the immunosuppressive effects of stress exposure, thereby preventing stress-induced encephalitis and mortality in WN-25-infected mice [13]. Venezuelan equine encephalomyelitis (VEE) is a significant disease affecting both humans and horses, caused by the VEE virus (VEEV), which is transmitted by mosquitoes. Outbreaks have occurred in northern South America from the 1920s to the 1970s, affecting thousands of individuals and equids. Mice serve as an experimental model, displaying agitation and increased movement followed by decreased mobility, paralysis, unconsciousness, and death upon VEE virus infection. Administration of melatonin protects VEE virus-infected mice by reducing virus levels in the brain and bloodstream, lowering mortality rates, delaying symptom onset, and extending survival time. Additionally, surviving mice treated with melatonin exhibit markedly elevated levels of VEE virus IgM antibodies [14]. Aleutian mink disease results from persistent infection with the Aleutian mink disease virus (AMDV). Minks in the advanced stages of the disease exhibit elevated levels of gamma globulins due to high concentrations of non-neutralizing AMDV antibodies, which are believed to cause lesions in the kidneys, liver, lungs, and arteries. Implants containing melatonin have been shown to decrease mortality in AMDV-infected minks [15]. All these data support a protector effect of melatonin against viral infections and justify the aim of this study, to know the role of melatonin in herpesvirus type 1 replication, as well as the mechanisms that regulate these effects.

## 2. Results

The inhibition of HSV-1 replication by melatonin was evaluated. The presence of the HSV-1 viral proteins in Vero cells 24 h after infection was detected by an immunofluorescence assay. A decrease in immunoreactivity in infected Vero cells treated with melatonin compared with no treated cells was observed. This result confirms the inhibitory role of the indole on HSV-1 replication (Figure 1).

Melatonin acts as a regulator of antioxidant enzyme activity. In this study, superoxide dismutase (SOD) and catalase (CAT) activities were analysed in the different experimental groups. While a significant increase in SOD activity was observed in Vero-infected cells, a decrease in CAT activity was found. These effects were partially prevented with MEL treatment, which decreases SOD activity and increases CAT activity (Figure 2a,b). On the other hand, the pro-oxidant action of the virus triggered the increase in total antioxidant activity (TAA) in an attempt to neutralize this action. A lower level of TAA was observed in infected cells treated with melatonin (Figure 2c). 

As viral proteins are mass-produced during herpesviral infection, the cell’s protein synthesis capacity, mainly at the endoplasmic reticulum level, suffers, leading to endoplasmic reticulum stress (ER stress), denoted by an increase in misfolded proteins that must be degraded by the proteosome in the cytoplasm. To confirm this, proteasome activity was measured. In infected cells, an increase in proteasome activity was observed, which was drastically reduced by Mel treatment (Figure 3a). When the production of misfolded proteins is exacerbated, the cell triggers, as a mechanism of repair and recovery of cellular homeostasis, the unfolded protein response (UPR). In infected cells treated with melatonin, increased phosphorylation of the α-subunit of eukaryotic initiation factor 2 (eIF2α) was observed, as well as an increase in the activating transcription factor 6-α (ATF6-α) 50/90 kDa ratio, indicating that this indolamine activated the UPR (Figure 3b,c).

A link between UPR and autophagy has been described, with autophagy being triggered by the activation of the eIF2α pathway belonging to UPR. A frequently used autophagy marker is LC3, which is converted from the cytosolic form LC3-I to the lower migrating form LC3-II upon autophagosome development. Western blot data indicate an increase in LC3-II/LC3-I ratio at 24 hpi in melatonin-treated infected cells (Figure 4a). A parallel increase in p62 protein expression and lysosomal-associated membrane protein 2A (LAMP-2A) gene expression was also observed in melatonin-treated infected cells, suggesting activation of macroautophagy and chaperone-mediated autophagy, respectively, both being markers of these autophagy subtypes (Figure 4b,c).

All these data together indicated that melatonin could trigger an autophagic process in HSV-1-infected cells. Autophagy as a survival mechanism and apoptosis as a mechanism of programmed cell death are independent mechanisms and clearly opposite in their cellular effect. To assess the status of apoptosis, the expression of pro-apoptotic BAX and anti-apoptotic B-cell lymphoma 2 (BCL-2) genes were studied. While BCL-2 showed no significant differences with either viral infection or melatonin treatment in any of the conditions studied, BAX showed a significant increase with viral infection, which was significantly reduced by melatonin treatment (Figure 5a,b).

## 3. Discussion

Melatonin has been defined as a “handyman” molecule able to prevent cell damage in several acute (sepsis, asphyxia in newborns) and chronic states (metabolic and neurodegenerative diseases, cancer, inflammation, aging) [16]. Thus, the efficacy of melatonin as a treatment of ocular diseases, cardiovascular diseases, sleep disturbances and several other pathologies, as well as a complementary treatment in anaesthesia, haemodialysis, in vitro fertilization and neonatal care, has been assessed [17]. Likewise, melatonin has been shown to reduce the toxicity and increase the efficacy of many drugs whose side effects are well documented [18]. The properties of melatonin as a potent antioxidant and inducer of antioxidant enzymes, as well as a regulator of certain biological processes such as reticulum endoplasmic stress, apoptosis and autophagy, explain its role. Viral infections which usually come up with oxidative stress increases and inflammatory injury are candidates to the beneficial actions of melatonin. Although several antiviral drugs to fight against viruses of clinical interest are available, new antiviral agents are needed to be developed to fight against viral infections with no treatment. The existence of resistant viruses is another cause to find these new antiviral drugs. This is the case of HSV-1, which infects mucosal epithelial cells, establishes life-long infections in sensory neurons and is the cause of symptoms ranging from genitourinary ulcers to encephalitis. The existence of resistant viruses supports the need to find new drugs. It has been reported that a formulation containing melatonin was able to regress a higher percentage of HSV-1 infection than using acyclovir [19]. The role of melatonin in HSV-1 replication was evaluated using a single clinical strain isolated from a patient, which is a limitation of this study. A decrease in antigen detection in Vero cells infected by HSV-1 was observed. A recent study has reported that melatonin reduces the immunohistochemical labelling of the capsid protein (VP60), as well as the VP60 mRNA expression in hepatocytes from RHDV-infected rabbits [20]. The mechanism altered by the viral infection and the role of melatonin to counteract these alterations were studied. It has been reported that oxidative stress is increased by herpesvirus infection across multiple tissues and species, and the effect size does not differ among different virus strains [21]. The increase in the total antioxidant activity detected in HSV-1-infected cells to counteract this oxidative stress supports this fact. So, the control of this pro-oxidant effect of HSV-1 is essential to the fight against the infection. Thus, the reported efficiency of several antioxidants, such as piperitenone oxide, vaticaffinol, ebselen, and resveratrol, among others, in reducing herpesvirus yield, is evident [21]. Furthermore, it has been shown that the expression of several antioxidant enzymes was differentially affected following a herpesviral infection [22,23]. Dysregulation of the expression of key antioxidant enzymes triggers a significant increase in oxidative stress, which is favourable for viral replication [21]. SOD and CAT are key enzymes in the first antioxidant defence pathway that must act in tandem to maintain their effectiveness in destroying free radicals. Thus, SOD destroys superoxide anions, transforming them into hydrogen peroxide which, thanks to the action of CAT, will be broken down into water and oxygen, thus undoing the oxidative pathway. However, if there is an imbalance in its action, such as an increase in SOD not matched by an equivalent in CAT activity, this situation will lead to an increase in H_2_O_2_ that is not neutralized by the second enzyme in the antioxidant pathway; this can therefore gradually be transformed into hydroxyl anion, which is much more harmful than the superoxide anion it was intended to destroy. At this juncture, unbalanced CAT SOD clearly functions as a pro-oxidant and not as an antioxidant, which was its essential function. Many pathological situations involve this imbalance of redox homeostasis with high levels of free radicals that potentiate and aggravate the initial damage [22,23]. This situation was reflected in the observed viral infection. Thus, increased SOD levels were observed in RHDV-infected cells and decreased CAT levels in mouse herpesvirus type 2-infected cells [23,24]. In turn, melatonin, one of the most potent antioxidants, was able to regulate the activity of antioxidant enzymes in HSV-1-infected Vero cells, decreasing SOD levels and increasing CAT levels to recover the antioxidant equilibrium. In fact, treatment with melatonin, which is much more multifunctional against oxidative stress than total TAA, having demonstrated its potency as a direct antioxidant [25] and its ability to induce the transcription of antioxidant enzymes [26], made the latter unnecessary, so that the increase in TAA observed in the viral infection was significantly reduced when the infected cells were treated with melatonin. Since, as we indicated, viruses require cellular machinery for the synthesis of their own viral proteins, the subsequent endoplasmic reticulum stress leads to an unexpected increase in misfolded proteins [27]. The proteasome system, which is the major pathway of intracellular protein degradation in eukaryotes, is a large, ATP-dependent, multisubunit protease; the mechanism involves cells physiologically regulating the concentration of particular or misfolded proteins by proteolysis. It has been reported that proteasome facilitates HSV-1 virus entry and that proteasome inhibitors, such as lactacystin and bortezomib, block early steps in HSV-1 infection [28].These results are probably related to the latency in the triggering of UPR that the maintenance of proteasome activity in the cell would cause, which would lead to a delay in the activation of the three cascades it includes and the cellular defence mechanisms involved. This hypothesis is supported by the results observed with melatonin treatment. The drastic reduction in proteasome activity in HSV-1-infected cells by treatment with Mel suggests that this indolamine may trigger this proteasome inhibitory effect and thus accelerate alertness in the infected cell. Such an alert, induced by a proteasome blockade, and thus early in melatonin-treated viral infection and later in viral culture, induces ER stress and consequently triggers UPR [29]. HSV-1 has been reported to try to inhibit this response to misfolded proteins to increase the time for the smooth utilization of the cell’s protein synthesis mechanisms [30], primarily at the level of the endoplasmic reticulum, essential for glycosylated protein synthesis. Indeed, HSV-1 has at least one glycosylated protein, glycoprotein G, which is essential for the HSV-1 invasion of neurons by increasing the amount of neurotrophic and neuroprotective proteins [31]. Due to the inability of the cytoplasm to glycosylate proteins, these can only be formed as such in the endoplasmic reticulum, so the virus is bound to replicate through this organelle, making it necessary to develop defence strategies against UPR.As a result, HSV-1 replication prevents ATF6 cleavage activation, as previously described in the early stages of infection [32], and melatonin is able to reverse this blockade. On the other hand, HSV-1 glycoprotein B (gB) prevents PERK activation and limits eIF2α phosphorylation [33,34], and these levels are increased by indolamine treatment. The effects of melatonin in promoting cell survival by modulating UPR at various levels have been described [7,35]. Our data show that, in the case of HSV-1-infected cells, melatonin increases ATF6 activity and eIF2α phosphorylation, at least partially reversing the mechanisms triggered by the virus to ensure viral protein translation. Activation of the ATF6 branch and the PERK pathway through phosphorylation of eIF2α has been suggested to promote autophagy [36,37]. HSV-1 has evolved several strategies to counteract autophagy: the dephosphorylation of eIF2α mediated by the HSV-1 protein ICP34.5 inhibits autophagy [38]. Herpesviruses lacking the γ34.5 gene trigger autophagy through activation of the eukaryotic translation initiation factor 2-kinase 2 (eIF2AK2)/protein kinase RNA-activated RNA (PKR) pathway [38,39]. The role of melatonin as a regulator of autophagy has been clearly established [40]. In our study, decreased expressions of LC3-I and LC3-II are observed in infected cells, indicating inhibition of the early steps of autophagy. This leads to a block of the autophagic pathway supported by the LC3-II/LC3-I ratio.Melatonin reverses this situation, favouring the development of autophagy, increasing LC3-II expression, unblocking the LC3-II/LC3-I ratio and increasing the levels of p62, a marker of viral aggresome disruption [41]. The reduction inLAMP-2A gene expression in infected cells, a marker of a complete block of the autophagic pathway, is also significantly increased by melatonin treatment, showing a complete recovery of the autophagic pathway. Autophagy and apoptosis are independent but clearly complementary mechanisms. Autophagy is the cell survival mechanism par excellence, so its inhibition reduces the chances of cell survival, favouring apoptosis [42]. Many viral genomes encode gene products that modulate apoptosis, either positively or negatively. Positively, the induction of apoptosis would directly contribute to the cytopathogenic effects of viruses, while the inhibition of apoptosis could prevent the premature death of infected cells, thus facilitating virus replication, propagation or persistence [43]. In HSV-1-infected cells, melatonin treatment causes a significant reduction in BAX expression, which is increased in untreated infected cells, indicating a suppression of apoptosis or an amelioration of cellular conditions that override programmed cell death. Melatonin has been shown in other situations, not caused by viral infection, to activate autophagy and suppress apoptosis, through oxidative stress-reduction mechanisms [44,45].

## 4. Materials and Methods

### 4.1. Cell Cultures and Viruses

Vero (African green monkey kidney) cells were cultured in Eagle’s MEM (Gibco BRL, Gaithersburg, MD, USA), supplemented with 10% foetal bovine serum (Sigma-Aldrich, Saint Louis, MO, USA) glutamine and gentamicin. An HSV-1 clinical isolate was propagated in Vero cells and titrated by plaque assay. 

### 4.2. HSV-1 Antigen Detection

Viral antigens were detected by immunofluorescence using specific monoclonal antibodies labelled with fluorescein isothiocyanate (IMAGEN Herpes Simplex Virus (HSV) Type 1 and 2; K610611-2 Thermo Fisher Scientific, Waltham, MA, USA).

After 24 h post-infection, the culture medium was removed and pure acetone (Sigma-Aldrich) was added. After incubating for 20 min to fix the cells, the acetone was removed and allowed to air dry. Monoclonal antibody was added at a 1/2 dilution and incubated at 37 °C for 30 min in an orbital shaker and in a humid chamber. After the incubation time had passed, the excess antibody was removed and two washes were carried out with PBS and one with water for 3 min each. The glass was removed from the Shell-vial, placed on a slide and visualized using the OLYMPUS BX-61 ultraviolet light microscope (Olympus, Tokio, Japan) where the staining of the infected cells wasobserved. In addition, images of different fields of vision were taken and analysed using the free software Image-J with the help of the Technical Scientific Services of the University of Oviedo.

### 4.3. Antioxidant Activity and Enzymatic Assays

Superoxide dismutase (SOD; EC1.15.1.1) activity was measured according to Martin et al. (1987) [46]. The enzyme inhibits the haematoxylin autooxidation to the coloured compound haematin. Results were expressed as SOD units per mg of protein. 

Catalase (CAT; EC 1.11.1.6) activity was assayed according to Lubinsky and Bewley (1979) [47] using H_2_O_2_ as substrate. Data are expressed as µmol of H_2_O_2_ consumed per mg of protein per minute. 

The protein amount was calculated using the Bradford method [48].

The total antioxidant activity was determined by the 2,2′-azino-bis (3-ethylbenzothiazoline-6-sulfonic acid) (ABTS)/H_2_O_2_/horseradish peroxidase (HRP) method [49], later modified [50]. In this method, H_2_O_2_ reacts with ABTS to form the ABTS radical in a reaction catalysed by HRP. ABTS has a characteristic absorbance spectrum with maxima at 414 and 730 nm. To test the samples, 5 µL of the sample or RIPA buffer used as a negative control was added and mixed with 245 µL of the ABTS radical. The difference between the initial and final absorbance value at 730 nm wasused as an index of antioxidant activity. The results were expressed in mg Tx/mg protein equivalents.

### 4.4. Immunoblotting

The cell lysates were mixed with 4×Laemmli buffer (Bio-Rad Laboratories, Inc., Hercules, CA, USA) and incubated for 5 min at 95°C to denature the proteins. Proteins were separated by 12% SDS-PAGE gel and transferred to polyvinylidene fluoride (PVDF) membranes using the mini-protean system (Bio-Rad Laboratories, USA) according to the manufacturer’s instructions. The membranes were blocked for 1 h at room temperature in 10% skimmed milk in TBS saline solution. The membranes were incubated overnight in a cold chamber at 4 °C with specific primary antibodies LC3 (1:1000) (Medical & Biological Lab, Tokyo, Japan), p62 (1:1000) (Abnova, Taipei, Taiwan), ATF-6 (1:500) (Santa Cruz Biotechnology, Santa Cruz, CA, USA) and eIF-2α (1:1000) (Cell Signaling Technology, Danvers, MA, USA) previously diluted in TBS buffer with 2% milk powder and 0.04% sodium azide.

The next day after removing the antibody, the membranes were washed three times for 10 min with TBS-t and incubated with the horseradish peroxidase (HRP)-conjugated secondary antibody diluted in TBS buffer with 2% milk powder for 1 h at room temperature. After the incubation period, the membranes were washed 3 times for 10 min each and developed using the HRP chemiluminescent substrate (Millipore Corporation, Darmstadt, Germany) according to the manufacturer’s instructions. Protein levels were quantitatively analysed using Image Studio Lite 3.1.4 software (LI-COR Bioscience, Lincoln, NE, USA) on the instrument (G BOX Syngene, Cambridge, UK). Since variations in the levels of housekeeping proteins (GAPDH, β-actin and α-tubulin) were found, the results were normalized to Ponceau S staining as a loading control [51].

### 4.5. Proteasome Activity

The proteasome activity was evaluated using a 20S proteasome activity assay kit (Chemicon, Merck Millipore, Billerica, MA, USA). The kit is based on the detection of the 7-amino-4-methylcoumarin (AMC) fluorophore following its cleavage of the labelled LLVY-AMC substrate by the chymotrypsin-like activity of the proteasome. Free AMC was detected using fluorimetric quantification (380/460 nm). Data were presented as arbitrary fluorescence units/mg protein.

### 4.6. Gene Expression Quantification

RNA was extracted from cell cultures using the TRI reagent (Sigma-Aldrich) and quantified using NANO DROP 3000 (Thermo Fisher Scientific, Waltham, MA, USA). Complementary DNA (cDNA) was synthesized using the High Capacity cDNA reverse transcription Kit (Applied Biosystems, Foster City, CA, USA) following the manufacturer’s protocols. Gene expression was analysed by quantitative real-time PCR (RT-qPCR) assays using a Light-Cycler DNA Master Plus SBYR Green I (Roche Diagnostics, Indianapolis, IN, USA) and the appropriate primers according to the Power SYBR Green PCR Master Mix protocol (4367659, Applied Biosystems). Primer pairs used for RT-PCR were forLAMP-2A, forward: 5′-GAAGTTCTTATATGTGCAACAAAGAGCAG-3′, reverse: 5′-CTAAAATTGCTCATATCCAGCATGATG-3′;BAX, forward: 5′-GCCAGCAAACTGGTGCTCA-3′, reverse: 5′-CCTGGTCTTGGATCCAGCC-3′; BCL-2, forward: 5′-TGCACACCTGGATCCAGGA-3′, reverse: 5′-CAGAGTCTTCAGAGACAGCCAGG-3′ and TBP, forward: 5′-TGCACAGGAGCCAAGAGTGAA-3′, reverse: 5′-CACATCACAGCTCCCCACCA-3′. The amplification protocol included the following stages: a holding stage for 10 min at 95 °C; a cycling stage for 45 cycles of 15 s at 95 °C, 10 s at 55 °C and 1 min at 60 °C; and finally, a melt curve stage for 15 s at 95 °C, 1 min at 60 °C and 15 min at 95 °C. cDNA samples were run, and the average cycle threshold (Ct) value at which each gene was detectable was calculated. The Ct of the TATA-binding protein (TBP) was used for normalization. The relative mRNA expression levels were calculated using the 2^−ΔΔCt^ method [52].

### 4.7. Statistical Analysis

All statistical analyses were performed using GraphPad Prism version 6.0 statistical software package for Windows (GraphPad Software, Inc., La Jolla, CA, USA). Data are presented as the mean ± standard deviation of the mean (SD). The normality of the data was analysed using the Kolmogorov–Smirnov test. Data were analysed using a two-way ANOVA and differences were analysed using the Bonferroni post hoc test. A *p* < 0.05 was considered statistically significant.

## 5. Conclusions

Melatonin, the most potent antioxidant known to date, shows in this study its ability not only to reduce the prevailing oxidative stress, but also to act effectively by unblocking the cellular security systems that the virus tries to neutralize. Melatonin’s ability to deactivate the proteosome and thus restore the response cascade to misfolded proteins is the key and, at the same time, is an essential step required by the cell to counteract infection. Thus, the recovery of the ATF6, eIF2α and PERK pathways allows the cell to activate multiple infection response pathways, some as efficient as autophagy in the fight against viral infections. This triggering of autophagy would render cell apoptosis unnecessary and indicate a victory of the cell over infection. Based on the results presented, melatonin is therefore not only an antioxidant that reduces oxidative stress, but a multifunctional molecule that enables the recovery of cellular defence, reducing viral blocking mechanisms and thus decisively influencing the success of the cellular antiviral fight. A scheme of how melatonin can regulate these processes due to its properties as an antioxidant and modulator of ER stress, autophagy and apoptosis is shown in Figure 6. This beneficial role of melatonin use as an additional therapeutic alternative against HSV-1 infection should be considered; further studies to corroborate this possibility should be performed.

## Figures and Tables

**Figure 1 ijms-25-04037-f001:**
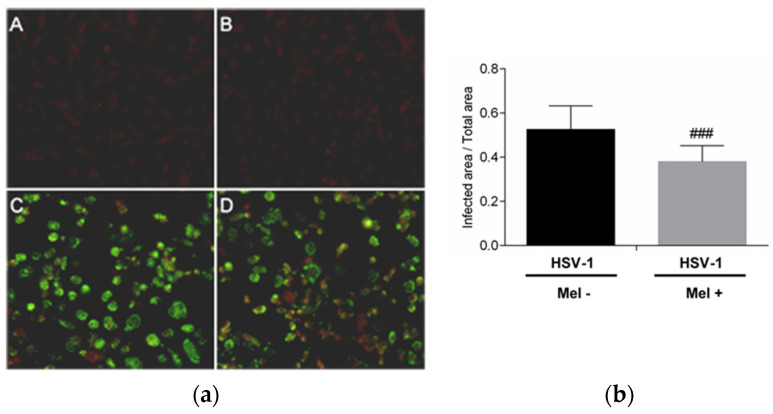
HSV-1 antigen detection by immunofluorescence in Vero cells. (**a**): Photomicrographs of sections of Vero cells taken from (**A**) Control, (**B**) Control + Mel (1 mM) (**C**) HSV-1, (**D**) HSV-1 + Mel (1 mM). Original magnification: 200×. (**b**): Image analysis of the area of antigen staining performed using the ImageJ software v3.91. Values are expressed as means (n = 10). ### *p* < 0.001 (HSV-1/HSV-1+Mel).

**Figure 2 ijms-25-04037-f002:**
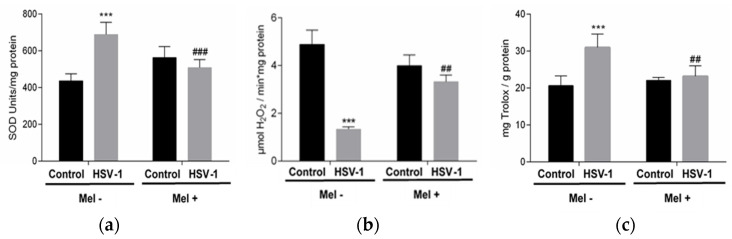
Effect of HSV-1 infection and melatonin treatment on antioxidant enzyme activity and total antioxidant activity. (**a**) Superoxide dismutase activity, (**b**) catalase activity and (**c**) total antioxidant activity. Values are expressed as means ± SEM (n = 6). *** *p* < 0.001, (Control/HSV-1), ### *p* < 0.001, (HSV-1/HSV-1+Mel), ## *p* < 0.01 (HSV/HSV-1+Mel).

**Figure 3 ijms-25-04037-f003:**
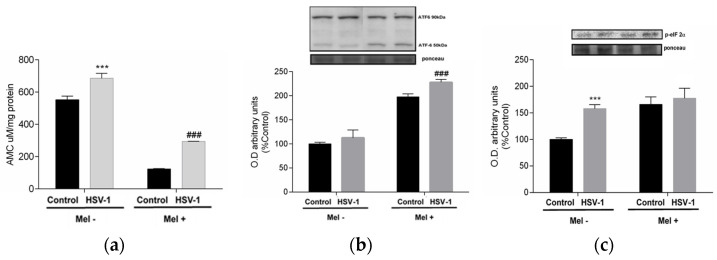
Effect of HSV-1 infection and melatonin treatment on proteasome and unfolded protein response (UPR). (**a**) Proteasome activity, (**b**) ratio ATF6-α 50/90 kDa expression and (**c**) p-eiF2-α expression. Values are expressed as means ± SEM (n = 6). *** *p* < 0.001, (Control/HSV-1), ### *p* < 0.001, (HSV/HSV-1+Mel).

**Figure 4 ijms-25-04037-f004:**
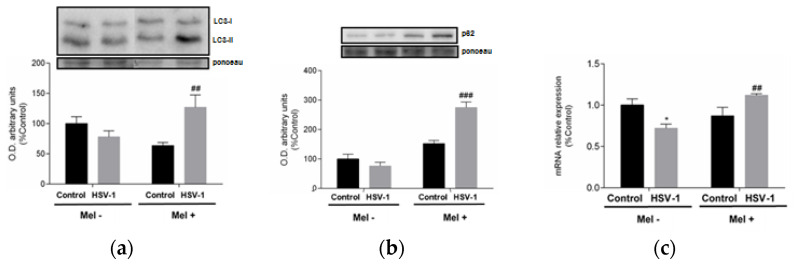
Effect of HSV-1 infection and melatonin treatment on autophagic response. (**a**) LC3-II/LC3-I ratio, (**b**) p62 protein expression and (**c**) LAMP-2A gene expression. Values are expressed as means ± SEM (n = 6). ## *p* < 0.01, (HSV-1/HSV-1+Mel), ### *p* < 0.001, (HSV-1/HSV-1+Mel), * *p* < 0.05, (Control/HSV-1).

**Figure 5 ijms-25-04037-f005:**
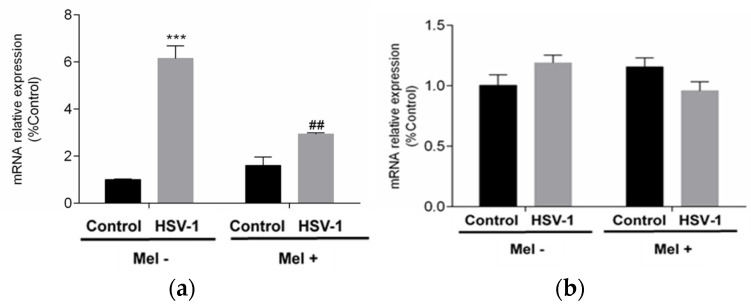
Effect of HSV-1 infection and melatonin treatment on apoptotic response. (**a**) BAX gene expression and (**b**) BCL-2 gene expression. Values are expressed as means ± SEM (n = 6). *** *p* < 0.001, (Control/HSV-1), ## *p* < 0.01, (HSV-1/HSV-1+Mel).

**Figure 6 ijms-25-04037-f006:**
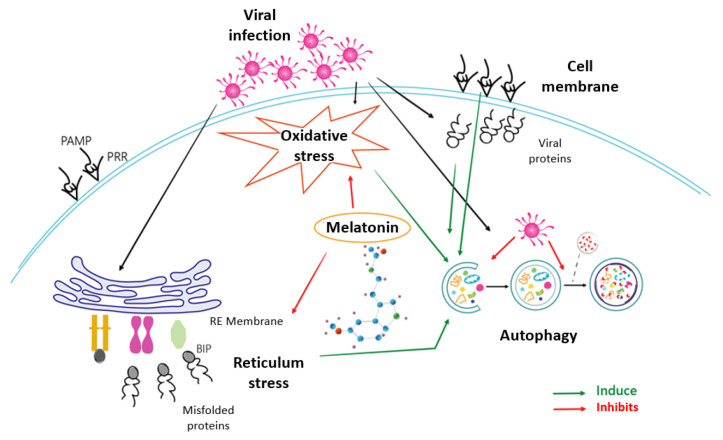
Mechanisms involved in the effect of melatonin on HSV-1 infection.

## Data Availability

The original contributions presented in the study are included in the article, further inquiries can be directed to the corresponding author/s.

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
