# Peer review of "Effect of Melatonin on Herpesvirus Type 1 Replication"

_ijms, 2024, doi:10.3390/ijms25074037_

Round 1

Reviewer 1 Report

Comments and Suggestions for Authors

The submitted article titled "Effect of Melatonin on Herpesvirus Type 1 Replication" demonstrates a well-articulated exploration into the potential inhibitory effects of melatonin on HSV-1 infection, a unique and intriguing research problem. The authors have effectively presented their findings, supported by clear graphical representations, and engaged in comprehensive discussion.

While the introduction provides a solid foundation, enhancing it with additional information about the role of melatonin in viral infections could further enrich the context.

The results are commendably presented and discussed in detail, demonstrating the authors' thorough understanding of the subject matter. However, the absence of a conclusive summary diminishes the overall impact. A well-structured conclusion would aid readers in comprehending the significance of the findings and their implications.

Furthermore, the provision of adequate references enhances the credibility of the article. In light of these observations, I strongly recommend the publication of this article in the International Journal of Molecular Sciences (IJMS), with the suggested minor enhancements incorporated.

Author Response

Reviewer 1

R1-The submitted article titled "Effect of Melatonin on Herpesvirus Type 1 Replication" demonstrates a well-articulated exploration into the potential inhibitory effects of melatonin on HSV-1 infection, a unique and intriguing research problem. The authors have effectively presented their findings, supported by clear graphical representations, and engaged in comprehensive discussion.

While the introduction provides a solid foundation, enhancing it with additional information about the role of melatonin in viral infections could further enrich the context.

A.-In the original manuscript, information about the role of melatonin in viral infections was explained in the discussion section. We agree with the reviewer that this information should be included in the introduction. So, in the new version the first reports where the role of melatonin, but not the molecular mechanisms involved were studied, are included in the introduction according to the reviewer`s suggestion. These evidences justify the aim of the article. The articles, where the molecular mechanisms  were studied are used to discuss our own results in the discussion section.

R.-The results are commendably presented and discussed in detail, demonstrating the authors' thorough understanding of the subject matter. However, the absence of a conclusive summary diminishes the overall impact. A well-structured conclusion would aid readers in comprehending the significance of the findings and their implications.

A.-We agree with the reviewer that a conclusive summary would improve the comprehension of the different mechanisms implicated in the action of melatonin. So, a conclusion section has been added to the end of the text.

R.-Furthermore, the provision of adequate references enhances the credibility of the article. In light of these observations, I strongly recommend the publication of this article in the International Journal of Molecular Sciences (IJMS), with the suggested minor enhancements incorporated.

Reviewer 2 Report

Comments and Suggestions for Authors

My maior caution is that the study was performed with only one virus strain (clinical, not reference). Some features are strain-dependent.

I have a comment on the sentence: This acute phase is followed by a period called latency before reactivation, which is associated with symptoms ranging from ulcers to encephalitis. This sentence needs improvement, the spectrum of symptoms/diseases caused by HSV-1 needs to be more clearly defined. In a separate sentence, the phenomenon of latency and reactivation can be pointed out.

Next sentence: Although several drugs against HSV-1 are available, new antiviral agents are needed to be developed due to the existence of resistant viruses.

To support this statement, I recommend that the authors list the drugs used to treat HSV-1 infection and briefly describe the problem of resistance (percentage of resistant isolates in immunocompetent and immunosuppressed patients).

This information can be included in the discussion section.

All abbreviations should be explained when used for the first time.

Figure 2 (b): the explanation "aa" should be given

 In vitro - should be written in italics

 I note the incorrect citation of Article 15

 Use subscript when writing hydrogen peroxide formula (should be corrected throughout the text)

Line 370: "Proteins were separated by 12% SDS-PAGE..."- ...better 12% SDS-PAGE gel....

Author Response

Reviewer 2

R2.- My maior caution is that the study was performed with only one virus strain (clinical, not reference). Some features are strain-dependent.

A.-Such as has been noted by the reviewer, the study was performed using a clinical isolate. We agree with the reviewer that some feature could be strain-dependent. However, similar studies reported in the scientific literature are performed using only one virus. In any case, this limitation has been included in the discussion section.

R2.-I have a comment on the sentence: This acute phase is followed by a period called latency before reactivation, which is associated with symptoms ranging from ulcers to encephalitis. This sentence needs improvement, the spectrum of symptoms/diseases caused by HSV-1 needs to be more clearly defined. In a separate sentence, the phenomenon of latency and reactivation can be pointed out.

A.-According to the reviewer`s suggestion the spectrum of symptons/diseases and the phenomenon of latency and reactivation have been defined more clearly.

R2.-Next sentence: Although several drugs against HSV-1 are available, new antiviral agents are needed to be developed due to the existence of resistant viruses. To support this statement, I recommend that the authors list the drugs used to treat HSV-1 infection and briefly describe the problem of resistance (percentage of resistant isolates in immunocompetent and immunosuppressed patients).This information can be included in the discussion section. Information about drugs used to treat HSV-1 infection and the problem of resistance has been included.

A.-According to the reviewer`s suggestion a brief list of the main drugs used to treat HSV-1 infection has been added to the text. On the other hand, data about prevalence of resistant isolates has also been added to the text.

R2.-All abbreviations should be explained when used for the first time.

A.-The abbreviations, such as TAA, ATF6, LAMP2A, BCL-2 were explained.R2.- 

R2.-Figure 2 (b): the explanation "aa" should be given

The character “aa” was a mistake and it has been deleted.

R2.- In vitro - should be written in italics.

A.-In vitro has been written in italics.

R2.- I note the incorrect citation of Article 15

A.-We have revised the citation of Article 15 and we do not know what the reviewer is refering to

R2.-Use subscript when writing hydrogen peroxide formula (should be corrected throughout the text)

 A.-H2O2 has been changed to H2O2.

R2.-Line 370: "Proteins were separated by 12% SDS-PAGE..."- ...better 12% SDS-PAGE gel....

A.-The word “gel” has been included.

Round 2

Reviewer 2 Report

Comments and Suggestions for Authors

I request that the authors highlight a section in the text (discussion) regarding the limitations of the study (regarding the use of a single virus strain). In response to the reviewer's comment, the authors replied that.  

R2. , "However, similar studies reported in the scientific literature are carried out using only a single virus. In any case, this limitation is discussed in the discussion section'.

The review is hampered by the lack of highlighted changes made to the revised manuscript.

I request that the authors carefully revise the entire text, removing unnecessary spaces, adding spaces, and writing the hydrogen peroxide formula correctly (e.g. in lines: 132, 133, 174, 247, 264, 274, 304, 321).

Author Response

Reviewer 2

R.-I request that the authors highlight a section in the text (discussion) regarding the limitations of the study (regarding the use of a single virus strain). In response to the reviewer's comment, the authors replied that. R2. , "However, similar studies reported in the scientific literature are carried out using only a single virus. In any case, this limitation is discussed in the discussion section'.

A.-  A sentence including this limitation has been included in the text (section discussion).

R.-The review is hampered by the lack of highlighted changes made to the revised manuscript.

A.- All changes made are highlighted in red.

R.-I request that the authors carefully revise the entire text, removing unnecessary spaces, adding spaces, and writing the hydrogen peroxide formula correctly (e.g. in lines: 132, 133, 174, 247, 264, 274, 304, 321).

A.- The text has been revised to remove unnecessary spaces and add spaces. Hydrogen peroxide formula was written correctly